# Comparative Review of the State of the Art in Research on the Porcine Epidemic Diarrhea Virus and SARS-CoV-2, Scope of Knowledge between Coronaviruses

**DOI:** 10.3390/v16020238

**Published:** 2024-02-02

**Authors:** Nora H. Rosas-Murrieta, Alan Rodríguez-Enríquez, Irma Herrera-Camacho, Lourdes Millán-Pérez-Peña, Gerardo Santos-López, José F. Rivera-Benítez

**Affiliations:** 1Centro de Química, Laboratorio de Bioquímica y Biología Molecular, Instituto de Ciencias, Benemérita Universidad Autónoma de Puebla, Puebla 72570, Mexico; alan.rodriguez.enriquez@gmail.com (A.R.-E.); irma.herrera@correo.buap.mx (I.H.-C.); lourdes.millan@correo.buap.mx (L.M.-P.-P.); 2Posgrado en Ciencias Químicas, Benemérita Universidad Autónoma de Puebla, Puebla 72570, Mexico; 3Centro de Investigación Biomédica de Oriente, Laboratorio de Biología Molecular y Virología, Instituto Mexicano del Seguro Social (IMSS), Metepec 74360, Mexico; gerardo.santos.lopez@gmail.com; 4Centro Nacional de Investigación Disciplinaria en Salud Animal e Inocuidad, Instituto Nacional de Investigaciones Forestales, Agrícolas y Pecuarias, Ciudad de México 38110, Mexico; expide@yahoo.com

**Keywords:** PEDV, SARS-CoV-2, coronaviruses, viral receptor, innate immune response, vaccines

## Abstract

This review presents comparative information corresponding to the progress in knowledge of some aspects of infection by the porcine epidemic diarrhea virus (PEDV) and severe acute respiratory syndrome coronavirus 2 (SARS-CoV-2) coronaviruses. PEDV is an alphacoronavirus of great economic importance due to the million-dollar losses it generates in the pig industry. PEDV has many similarities to the SARS-CoV-2 betacoronavirus that causes COVID-19 disease. This review presents possible scenarios for SARS-CoV-2 based on the collected literature on PEDV and the tools or strategies currently developed for SARS-CoV-2 that would be useful in PEDV research. The speed of the study of SARS-CoV-2 and the generation of strategies to control the pandemic was possible due to the knowledge derived from infections caused by other human coronaviruses such as severe acute respiratory syndrome (SARS) and middle east respiratory syndrome (MERS). Therefore, from the information obtained from several coronaviruses, the current and future behavior of SARS-CoV-2 could be inferred and, with the large amount of information on the virus that causes COVID-19, the study of PEDV could be improved and probably that of new emerging and re-emerging coronaviruses.

## 1. Introduction

Every year, the pork industry is severely affected by viral infections, with porcine epidemic diarrhea (PED) being one of the highly contagious diseases that most concerns producers. The disease is caused by PEDV. There are various strategies to control infection outbreaks on farms such as biosecurity programs, appropriate zootechnical management practices, and vaccines. The occurrence of mutations in the viral genome motivates the continuous search for new alternatives to control PED for producers. In this review, comparative information is presented on some aspects of porcine epidemic diarrhea and COVID-19, two diseases caused by coronaviruses, PEDV and SARS-CoV-2, respectively.

PEDV causes vomiting, dehydration, diarrhea, and weight loss in pigs, with mortality rates of up to 100% in piglets [1], while SARS-CoV-2 causes fever, dry cough, dyspnea, pulmonary pneumonia, edema, acute respiratory distress syndrome, and, in severe cases, there is multiple organ failure [2,3], associated with high morbidity and mortality. PEDV belongs to the order Nidovirales, family Coronaviridae, subfamily Coronavirinae, genus Alphacoronavirus. SARS-CoV-2 belongs to the same order, family, and subfamily, but to the Betacoronavirus genus. Both viruses contain linear capped and polyadenylated ssRNA (+) genomes. The linear genome of PEDV (~28 Kb) encodes 16 non-structural proteins (NSP1–16) and 5 structural proteins, S, M, N, E, and ORF3 [1]. The linear genome of SARS-CoV-2 (~29.8 kb) encodes sixteen non-structural proteins (NSP1–16), four structural proteins, S, M, N, and E, and accessory proteins such as ORF 3a, 3b, 6., 7a, 7b, 8, 9b, 10, and 14 [3], as shown in Figure 1.

The impact of the COVID-19 pandemic on all areas of human life motivated the international scientific community to decipher the underlying mechanisms of the infection caused by SARS-CoV-2, generating a large amount of information since the end of 2019 and early 2020. Part of the information published about SARS-CoV-2 is consistent with that described for other coronaviruses such as PEDV, which has been studied for more than 40 years. The first reported occurrence of PEDV cases occurred in 1977 in England and in Asia in 1981 [1]. However, information on PEDV infection is still limited. PEDV and SARS-CoV-2 share several characteristics, from the general structure of the virion to the cellular response induced in the host. Knowledge of infection by PEDV and other coronaviruses allowed the initial approach for the characterization of SARS-CoV-2 replication and the development of some control strategies. It is possible that the extensive knowledge of SARS-CoV-2 could be applied in the characterization of emerging or re-emerging PEDV strains and in the creation of new control strategies on pig farms. Table 1 and Table 2 show comparative information for both viruses to expose the potential scope of data between the coronaviruses.

## 2. Transmission Route

One of the first aspects to be studied in any emerging or re-emerging virus is the transmission route. Currently, these viruses are known to have similar transmission routes. The fecal–oral route is the main route of transmission of PEDV. It is also known that the virus can spread through air currents [4], and up to 3.5 × 10^8^ RNA copies/m^3^ of PEDV [5] have been detected within 10 km of an infected farm [6]. Furthermore, the presence of the viral genome has been demonstrated in the nasal cavity of pigs that did not have direct or indirect contact with pigs infected with PEDV [7]. In newborn piglets, PEDV initially replicates in intranasal epithelial cells, which transfer the virus to CD3^+^ T lymphocytes, which reach the intestine through the blood, releasing the virus to the enterocytes for replication [8].

On the other hand, transmission of SARS-CoV-2 occurs by direct contact through coughing, sneezing, contact with oral, nasal, and ocular mucosa, and aerosols [9,10]. SARS-CoV-2 is adsorbed on dust and transported over long distances from a contaminated environment, and the virus is viable in aerosols for up to 3 h [11]. At the beginning of the COVID-19 pandemic, it would have been useful to consider that SARS-CoV-2 shares the air route of transmission with PEDV. Therefore, the implementation of the use of masks from the early stages of the COVID-19 pandemic would have reduced the contagion curve throughout the planet.

As already mentioned, it is known that the main route of transmission of PEDV is the fecal–oral route, a fact that was reported late in SARS-CoV-2 infection [12,13,14,15]. In a severe case of COVID-19, viral RNA was detected in the esophagus, stomach, and rectum [16]. The SARS-CoV-2 virus infects and replicates efficiently in human intestinal epithelial cells [17,18]. In some patients infected with SARS-CoV-2, characteristic symptoms of gastrointestinal diseases such as diarrhea, anorexia, and nausea occurred [19,20]. Some studies have reported the development of diarrhea in between 2 and 50% of COVID-19 cases [21,22,23].

On the other hand, the isolation of SARS-CoV-2 from anal samples at late stages of viral infection suggests that the infection in intestinal cells is longer than in the oral cavity [24,25,26]. This information implies the possibility that a patient diagnosed as negative for SARS-CoV-2 from an oral sample but positive in an anal sample could be classified as a false negative [27,28], potentially being a source of viral transmission.

During the first hours of infection, efficient replication of PEDV in enterochromaffin cells of the intestinal submucosa has been reported, then the virus infects the middle region of the jejunum and ileum and subsequently the proximal and distal jejunum as well as the duodenum [29]. Atrophic enteritis manifests between days 3 and 5 after infection [30].

SARS-CoV-2 infection of enterocytes produces acute necrosis and exfoliation of the lamina propria, causing vomiting, diarrhea, and malabsorption, which are characteristic intestinal symptoms of PEDV infection [29,30]. If the use of this information in the spread of COVID-19 had been considered, an intense campaign of hygiene measures among the population would have reduced the number of infected individuals. It is necessary to elucidate the molecular mechanism by which SARS-CoV-2 causes the gastrointestinal disease in humans to propose new strategies focused on reducing the symptoms and severity of the COVID-19 disease.

PEDV has also been detected in lung tissue, mainly in epithelial cells and alveolar macrophages [31]. PEDV infects dendritic cells derived from porcine monocytes, although it does not replicate in this cell type [32]. Overcoming the epithelial/endothelial barrier could be a shared characteristic between coronaviruses. Apparently PEDV uses dendritic cells to cross the epithelial barrier, evade the antiviral immune response, and reach the submucosal layer [33,34].

To date, the mechanism of SARS-CoV-2 to overcome the alveolar barrier and achieve efficient viral replication in the respiratory system is not known in detail. In severe cases of COVID-19, alteration of pulmonary epithelial and endothelial barriers has been observed. In an in vitro system, it was shown that SARS-CoV-2 efficiently infects epithelial cells with high viral loads, inducing the interferon response but does not infect the adjacent endothelial layer. However, in prolonged infections, both types of cells are damaged, and the barrier is modified to favor the spread of the virus [35]. This suggests that transit through the epithelial/endothelial barrier could be a possible target for the design of strategies that interrupt coronavirus viral infection.

Alteration of epithelial and endothelial barriers is a characteristic associated with the development of severe cases of COVID-19, which generally occurs in patients with underlying diseases such as diabetes or aging. It has been proposed that glucagon-like peptide 1 (GLP-1) signaling activates barrier-promoting processes that counteract the proinflammatory mechanisms induced by SARS-CoV-2, making the signaling pathway a potential therapeutic target [36]. This fact will need to be addressed in PEDV infections.

On the other hand, the detection of PEDV genetic material in semen of infected pigs raises the possibility of considering another route of disease dissemination [37,38]. In patients with COVID-19, the virus has been reported in the testicle and epididymis, affecting the production of sperm and testosterone, with a decrease in sperm quality, as well as the occurrence of orchitis in the pediatric population and in adults [39,40]. However, the probability of SARS-CoV-2 being transmitted through semen is very small [41].

The use of information on the known transmission mechanisms of PEDV at the beginning of the COVID-19 pandemic would have led to the immediate implementation of control measures such as the use of masks, as well as particle containment methods or disinfection of aerial sprays used in farms infected by PEDV. Furthermore, more research is required to understand in detail whether SARS-CoV-2 uses mechanisms like PEDV to reach intestinal tissue. This knowledge would be useful to propose new strategies to control infection by both PEDV and SARS-CoV-2.

## 3. Tropism and Receptors

S protein is a type 1 membrane glycoprotein, which forms projections on the viral envelope with a crown-like appearance in electron micrographs. S protein is the protein responsible for the tropism of the PEDV and SARS-CoV-2 viruses [42]. S protein of PEDV and SARS-CoV-2 contains an N-terminal extraviral region, a transmembrane domain, and a small intraviral region. The ectodomain of S protein consists of a receptor-binding subunit, termed S1, which contains the receptor-binding domain (RBD) to engage the host cell receptor, thereby determining virus cell tropism and pathogenicity. The S2 domain contains a transmembrane region, heptad repeat regions, and the fusion peptide, which promotes the fusion of viral envelope and cellular membrane [43,44]. The S1 domain is organized into the N-terminal domain (S1-NTD) and the C-terminal domain (S1-CTD), and both are functional as RBDs.

In 2007, it was recognized that alanyl aminopeptidase (AAP) or aminopeptidase N (AP-N, APN, or CD13) is the receptor for PEDV [45,46,47]. APN is a protein that is highly expressed in the apical membrane of enterocytes [48], which explains the tropism of PEDV for this tissue and its pathology. APN interacts with the viral S protein, which also binds sialic acid. PEDV has been shown to interact with APN through the S1-CTD domain of the S protein [49], and the enzymatic activity of APN promotes PEDV replication [50].

However, it has also been reported that MDCK cells overexpressing APN are not susceptible to PEDV. Under the working conditions used, the PEDV S protein does not bind to membrane APN or its soluble form. Nor did preincubation of protein S with the soluble form of the enzyme have any effect on PEDV infection. The same work reported the use of CRISPR/Cas9 to suppress APN expression in PEDV-susceptible Huh7 and HeLa cell lines and confirmed that the transmembrane enzyme APN is not essential for PEDV entry into cells [51,52,53]. However, other groups have reported that the APN enzyme allows PEDV to spread laterally in intestinal epithelial cells [45,54].

Interestingly, other working groups have reported that PEDV can bind to other receptors since infection with the virus is successful in APN-knockout pigs [52,53,55], as well as in cells of human, monkey, and bat origin [56,57], suggesting the existence of a receptor for PEDV and its possible zoonotic transmission. In piglets, the S1 domain of the viral S protein was shown to interact with the extracellular region of transferrin receptor 1 (TfR1), highly expressed in the apical region of intestinal cells [58].

On the other hand, it is known that some carbohydrates can serve as co-receptors for PEDV. The S1-NTD domain of the S protein binds to N-acetylneuraminic acid (Neu5Ac or NANA) (higher affinity), 5-N-glycolyl neuraminic acid, or Neu5Gc and mucins [55,59].

It has also been reported that cholesterol is a molecule required for PEDV infection; pharmacological sequestration of this lipid effectively blocks both virus binding and internalization [60]. Furthermore, occludin and αvβ3 integrin enhance PEDV entry and replication [61,62]. It was recently demonstrated in the Vero cell line that the death receptor DR5 facilitates PEDV entry and replication by regulating caspase-8-dependent apoptosis [63].

The receptor of the SARS-CoV-2 virus is angiotensin-converting enzyme 2 (ACE2) that interacts with the S1-CTD domain of the S viral protein [64,65]. ACE2 is highly expressed in the tongue, oral mucosa, salivary glands, and small intestine. The detection of SARS-CoV-2 in intestinal tissue and the clinical characteristics of a gastrointestinal disease observed in SARS-CoV-2-infected patients can be explained by the expression of ACE2 [66]. Single-cell RNAseq analysis has shown that ACE2 is highly expressed in enterocytes of the ileum, colon, and stomach [67,68,69]. SARS-CoV-2 virus has also been observed in the cytoplasm of gastric, duodenal, and rectal glandular epithelial cells [14].

High levels of expression of ACE2 have also been reported in testicles, kidneys, heart, thyroid, adipose tissue, enterocytes, bladder, cardiomyocytes, male reproductive cells, placental trophoblasts, ductal cells, and ocular tissue [67,68]. In lung tissue, high expression has been reported in bronchial transient secretory cells, in nasal and bronchial epithelium, and in alveolar type II epithelial cells [70,71]. Interestingly, those tissues with medium levels of expression are the colon, liver, bladder, adrenal gland, and some regions of the lung. Among the tissues with lower levels of expression are the spleen, bone marrow, brain, blood vessels, and muscles [72,73,74]. Despite the different levels of ACE2 expression in the human body, its presence in various tissues could explain the multiorgan damage present in some patients infected by the virus [75,76,77].

APN and ACE2 are peptidases, a distinctive feature of the receptors of different coronaviruses. It is known that SARS-CoV-2 requires the participation of proteases such as transmembrane serine protease 2 (TMPRSS2), serine protease 4 (TMPRSS4), furin, or other enzymes for viral entry into cells [18,19]. For the propagation of PEDV in cell cultures, it is necessary to supplement the medium with trypsin [78]. Gastric and pancreatic proteases, in addition to proteases expressed by intestinal epithelial cells, can carry out this proteolytic process in PEDV infection; this activity could be added to that of ACE2 to explain the tropism of the virus to the enteric tract. In the receptor-binding state, S proteins of PEDV and SARS-CoV-2 require proteolytic processing for the fusion process [79,80,81,82]. TMPRSS2 protease performs proteolytic processing on the S protein of SARS-CoV-2 [82]. The TMPRSS2 protease is highly expressed in intestinal cells as well as in the salivary glands (ductal, acinar, and myoepithelial cells) and the tongue (the layer of spinous base cells, the stratum corneum, and the epithelial surface) [72,73].

Considering that SARS-CoV-2 requires ACE2 and TMPRSS2 proteins to initiate the viral replication cycle. It is possible to point out that all human tissues that express both proteins could be susceptible to the viral infection [83]. In ACE2 enterocytes, it was shown that the successful replication of SARS-CoV-2 requires the enzymes TMPRSS2 and TMPRSS4, which facilitate the fusogenic activity of the viral S protein and the entry of the virus into host cells in vitro [18,84].

In relation to PEDV, the successful generation of the Vero/TMPRSS2 and Vero/MSPL cell lines, which constitutively express TMPRSS2 and MSPL (trypsin-like serine protease), increased the viral titer and isolation of PEDV in vitro [85,86].

Despite the large difference in the depth of knowledge about receptors between coronaviruses, it is possible to share receptor information between SARS-CoV-2 and PEDV. However, more research is required to know the extent of information between both viruses. For example, the molecular mechanism of SARS-CoV-2 arrival in the intestinal tissue must be deciphered and such information compared with what has been reported for PEDV. The findings could lead to the design of new strategies that help reduce the symptoms and severity of PEDV and SARS-CoV-2 infections.

## 4. Virus Classification and Variants

PEDV is organized into two genogroups, G1 and G2, which can be divided into G1a, G1b, G2a, and G2b. Genogroup G1a includes classical strains such as the prototype strain CV777. Genogroup G1b contains strains isolated in Asia that present deletions in different genes, mainly those encoding the NSP3 and ORF3 proteins [1]. Genogroup G2 differs from G1 mainly by the amino acid changes in the NTD domain of S protein. In genogroup G2a, there are circulating Asian strains reported as re-emerging strains of PEDV. In the G2b genogroup, strains are found circulating in the United States and other parts of the world [87,88].

The high mutation rate of the *S* gene [89,90,91] has allowed its use in the classification of PEDV strains. From the *S* gene sequences, there are well-described genotypes in the United States called G1b (S-INDEL) and G2b (American prototype or non-S-INDEL) [92]. US S-INDEL PEDV is a variant strain that emerged in 2014 in the United States and contains insertions and deletions (INDELs) in the *S* gene [93,94,95]. The US non-S-INDEL PEDV is genetically like pathogenic strains that emerged in China in 2010 [96]. PEDV S-INDEL is less pathogenic than non-S-INDEL PEDV in piglets [95,96,97].

Additionally, the analysis of the complete genome of several PEDV strains has identified hypervariable regions in ORF1 that affect the proteins NSP2, NSP3, the S1 and S2 domain of the S protein, ORF3, and the N protein [97,98]. On the other hand, phylogenetic inferences of PEDV using the N and ORF3 proteins have grouped the strains into two large groups, G1 and G2. G2 is composed of three subgroups G2-1, G2-2, and G2-3 [99,100].

Several phylogenetic analyses have been performed with genomes of different SARS-CoV-2 strains isolated from different patients around the world and at different times. However, due to the magnitude of the pandemic, classification of SARS-CoV-2 strains has been challenging [101].

The analysis of 12,343 genomes of SARS-CoV-2 strains deposited in the Global Initiative on Sharing Avian Influenza Data (GISAID) found more than 1000 mutations, of which 131 have a frequency greater than 10%. Among the most relevant are the D614G mutation in the S protein and P4715L in the NSP12 protein [102], which is important for viral RNA replication. The D614G mutation has been found widely distributed in Europe and has been associated with increased virulence and mortality [103,104]. The P4715L mutation appears to be associated with increased mortality, although the functional significance of the mutation has not been clarified.

A phylogenetic classification system (Pango) based on SARS-CoV-2 lineages has also been developed based on several factors for the identification of the strains that contribute most to the active spread of the virus [105,106,107]. There are two main lineages, named A and B, in the phylogeny of SARS-CoV-2. Lineage A viruses include the Wuhan/WHO4/2020 sequence registered in January 2020 that share some similarity in the ORF1ab and ORF8 sequence with bat viruses. The B lineage includes the Wuhan-Hu-1 strain recorded in December 2019, which is different in the ORFs mentioned above [105].

The complex scientific names of SARS-CoV-2 variants led the WHO to create a simpler naming system for the most important ones. Starting in May 2021, the WHO began using letters of the Greek alphabet [108]. This facilitated the monitoring and dissemination of information widely to the non-specialized public. Therefore, the B.1.1.7 variant detected in September 2020 in the United Kingdom became the Alpha variant, while B.1.351, B.1.1.28.1 (P1), B.1.617.2, and B.1.427 were named Beta, Gamma, Delta, and Epsilon, respectively [109,110]. The number of concerning new variants continued to grow until reaching the Omicron variant, which was the last to receive this type of name. Today, most circulating variants are descended from Omicron [111].

Nextstrain is a project for real-time monitoring of the evolution of pathogens based on genome data and has evaluated the phylogenetic relationship of clades between SARS-CoV-2 strains, organizing them into three large groups. The first group is made up of clades 19A and 19B, which include emerging variants in Wuhan and the first ones related to the pandemic outbreak. The second group includes clade 20A, which covers variants that derive from group 19A and the dominant variants from the European outbreak in March 2020 and variants spread globally. The latter group is made up of clades 20B and 20C, which are large subclades genetically distinct from clade 20A [112].

On the other hand, on the GISAID platform (https://www.gisaid.org/) (consulted on 16 January 2024) [113,114] until January 2024, almost 16,500,000 genome sequence submissions of virus sequences have been reported. GISAID uses a sophisticated system to classify genomes by region, date, whole genome, or gene. The platform employs a nomenclature system for the main clades developed by Sebastian Maurer-Stroh et al., which considers the main clades, based on marker gene mutations, to organize them into six phylogenetic groupings. Initially, the clades were organized into groups S and L. Later, L was divided into groups V and G and then group G was divided into GH and GR [115,116,117,118].

The classification systems of SARS-CoV-2 could be used for the phylogenetic classification of PEDV and other coronaviruses, which would facilitate almost real-time monitoring of the genetic variants.

### Recombination

Different studies have reported the occurrence of recombination events between PEDV strains [119,120,121,122]. Such events have caused the phylogenetic classification to be extended to an additional group, G2c [123,124], which is made up of strains whose genomes have experienced recombination with strains from other genogroups. Interestingly, recombination between PEDV strains of transmissible gastroenteritis virus or TGEV has been proven [125,126]. The virulent strain called CN/Liaoning 25/2018 is the product of recombination between a highly virulent and an attenuated strain of PEDV [127]. The great variability of the PEDV genome can be attributed to recombination events in the S1 domain of PEDV, which would cause differences between circulating strains and mutability in the S protein [96,128].

Recombination events in SARS-CoV-2 have been documented on multiple occasions [129]. Recombinant viruses are nominally identified using an X followed by sequential letters, resulting in designations like XA, XB, XC, XAA, XAB [107,130]. The first recognized recombinant variant, XA, emerged in the United Kingdom at the end of 2020 and was the result of viruses from the B.1.1.7 and B.1.177 lineages [131]. One of the first recombinant viruses that gained a large territorial presence was the so-called XB, from the recombination of the B.1.634 and B.1.631 lineages, which emerged after high transmission in North America in 2021 [132]. In 2022 and beyond, the identification and genetic characterization of various recombinant variants increased, with some gaining attention in the general media, such as Deltacron (formed from Delta and Omicron variants, e.g., XD, XF, or XS), Gryphon (characterized as XBB), and Hippogryph (characterized as XBB.1), derived from Gryphon, or Kraken (characterized as XBB.1.5), derived from Hippogryph [133,134,135]. Notably, the latest recombinant viruses mentioned correspond to Omicron subvariants. By the end of 2023, Omicron subvariants, such as BA.2.86 (Pirola), and recombinant descendants of XBB maintain a significant population presence [136,137]. The genetic variation of SARS-CoV-2 continues to evolve, emphasizing the necessity for ongoing global, regional, and country-specific monitoring.

The possibility of recombination events between SARS-CoV-2 strains or with other human coronaviruses has been raised, as well as changes in the PEDV S protein, due to mutations or recombination events, the appearance of re-emerging strains, and the low effectiveness of control strategies, including vaccination, which have made it difficult to control the disease.

## 5. Immune Response

The generation of efficient strategies for the control of diseases caused by viruses depends on the knowledge of the molecular mechanisms of viral infection, with the level of interaction between virus replication and the antiviral cellular response being notable. PEDV infection suppresses the innate immune response using structural and non-structural viral proteins to target the adapters of innate immune pathways to inhibit type I IFN production.

Coronavirus non-structural protein 7 (NSP7) acts with NSP8 in the integration of an RNA primase required for viral RNA synthesis. NSP7 antagonizes the interferon beta (IFN-β) production by the competition with the PP1 phosphatase to bind to MDA5. NSP7 also affects the activation of transcription factors interferon regulatory factor 3 (IRF3) and nuclear factor-kappa B (NF-κB) [138]. PEDV NSP7 prevented the nuclear translocation of ISGF3, by interacting with the DNA-binding domain of STAT1/STAT2, to avoid the interaction with karyopherin α1 [139].

For type I IFN production, interferon regulatory factor 7 (IRF7) must be phosphorylated by TBK1 and IKKε. PEDV M protein interacts with IRF7 to suppress its phosphorylation and dimerization and to decrease type I IFN expression [140]. PEDV nucleocapsid (N) protein negatively regulates histone deacetylase 1 (HDAC1) expression and affects the Sp1-HDAC1-STAT1 signaling axis. Therefore, it inhibits the transcription of some ISGs such as OAS1 or ISG15 and favors viral replication [141,142]. Using a porcine intestinal epithelial cell line, it was demonstrated that PEDV antagonizes the antiviral response by the type III IFN through NSP1, NSP 3, NSP 5, NSP 8, NSP14, NSP15, NSP16, open reading frame 3 (ORF3), E, M, and N. NSP1 is the viral factor that inhibits the nuclear translocation of IRF1 [143].

A distinct antagonistic strategy of PEDV to circumvent the host antiviral response targets FBXW7, an innate antiviral factor capable of enhancing the expression of RIG-I and TBK1 and to induce some ISGs. In pig intestine, the viral protein NSP2 interacts with FBXW7 to promote the cellular factor degradation by the K48-linked ubiquitin-proteasome pathway [144].

The PEDV envelope protein (E) is a viroporin useful in the budding and assembly stages of the virus. It is known to induce endoplasmic reticulum stress (ERS) through phosphorylation of PERK/eIF2α components [145]. Also, PEDV E protein inhibits the translocation of IRF3 and suppresses retinoic-acid-inducible gene I (RIG-I)-mediated signaling to block IFN-β production [146].

PEDV ORF3 down-regulates the expression of signal molecules in the RLRs-mediated pathway, counteracting the transcription of IFN-β and ISG mRNAs [147]. It was recently described that KLF16, a member of the Krüppel-like factors (KLFs), inhibits PEDV viral replication by up-regulating the expression of IFNs and promoting the host antiviral innate immune response via the TRAF6-pTBK1-pIRF3 pathway [148].

SARS-CoV-2 evades the host immune system through several mechanisms conserved in other coronaviruses [149]. In vitro analysis of genes co-expressed with angiotensin-converting enzyme 2 identified to IL6, IL1, IL20, IL33, and IFI16 as regulators of the IFN response [150]. SARS-CoV-2 suppresses IFN-β production, interferes with the TANK-binding kinase 1 (TBK1)/interferon regulatory factor 3 (IRF3) activation axis, and prevents STAT1 and STAT2 phosphorylation.

SARS-CoV-2 proteins that have been associated with blocking the immune system are NSP1, NSP6, NSP8, NSP12, NSP13, NSP14, NSP15, open reading frame (ORF) 3a, ORF6, ORF8, ORF9b, ORF10, and membrane protein [151]. It has been described that the N protein of SARS-CoV-2 represses IFN-β production through the RIG-I protein [152]. The SARS-CoV-2 proteins NSP3, NSP5, NSP7, NSP8, M, and ORF9b suppress the production of type I and III interferons (IFNs) by altering the RIG-I/MDA5-MAVS signalosome, which reduces the phosphorylation of IRF3 and its nuclear translocation. Nsp7 also blocks the TLR3-TRIF and cGAS-STING signaling pathways [153,154,155,156].

In the most severe forms of COVID-19, there is an exacerbated production of proinflammatory cytokines and extensive cellular infiltrates in the respiratory tract. N protein induces the expression of proinflammatory cytokines by activating NF-κB signaling, the NLRP3 inflammasome, and finally producing a cytokine storm [157,158]. SARS-CoV-2 replication induces a delayed IFN response in lung epithelial cells. MDA5 and LGP2 regulate IFN induction in response to SARS-CoV-2 infection. Furthermore, IRF3, IRF5, and NF-κB/p65 are the key transcription factors that regulate the IFN response during SARS-CoV-2 infection [159].

Finally, the innate immune response executed by IFNs and induced by both coronaviruses and the ability of the viruses to block the antiviral cellular response have been studied in sufficient depth. On the other hand, the rapid characterization of the relationship between SARS-CoV-2 replication and the suppression of the host immune response has allowed us to understand its role in the pathogenesis of COVID-19. It is possible that this knowledge of SARS-CoV-2 will be extended to PEDV to improve understanding of the contribution of viral proteins in the pathogenesis of DEP. Molecular knowledge of the interaction between the viral proteins of coronaviruses and the innate immune response is vital to design possible viral control strategies.

## 6. Antibodies

Based on the knowledge of the structure and immunogenic properties of some SARS-CoV-2 viral proteins, the development of therapeutic antibodies was proposed through the phage display technique [160], using antibody gene libraries from donors without prior contact with infectious agents. such as hepatitis C, influenza, and HIV viruses, among others. Antibodies against SARS-CoV-2 were obtained from a library generated pre-pandemic from a healthy donor. These antibodies are capable of binding to the S1 region of S protein and RBD and have neutralizing activity. This demonstrates that universal libraries from healthy human donors offer a great alternative to generate therapeutic antibodies quickly and independently of the availability of biological material from patients and can be a quick solution in pandemics or the emergence of a new class of microorganism that represents a health risk to the population [161].

Understanding the characteristics and neutralizing potential of therapeutic antibodies against SARS-CoV-2 opens the possibility for the development of new therapy against COVID-19 and the rational design of next-generation vaccines. The possibility of having a library of pig antibodies using the same or a similar technique would offer a tool to confront new PEDV variants or new viruses that affect pigs.

## 7. Vaccines

Currently, different vaccines against coronavirus infections have been formulated based on mRNA technology, DNA, viral vectors, and recombinant proteins as well as attenuated and inactivated strains. The development of vaccines against SARS-CoV-2 has been dizzying, and more than 100 vaccine candidates have been reported [162].

Pfizer-BioNTech’s BNT162b1 and Moderna’s mRNA-1273 are two of the most widely used vaccines against SARS-CoV-2. Both vaccines content a nucleoside-modified mRNA encoding the viral spike within lipid nanoparticles for delivery into cells [163,164]. The company Inovio Pharmaceuticals generated a DNA vaccine called INO-4800, which includes the *S* gene cloned in the pGX9501 vector [165]. The company CanSino Biological Inc. has developed the vaccine called Ad5-nCoV that uses the adenovirus Ad5 vector, which contains the gene encoding the S protein [166]. The same technology is applied by Janssen Vaccines & Prevention, which offers the adenovirus-based Ad26 vaccine that includes the *S* gene [167]. Another vaccine that uses this technology is called ChAdOx1-nCoV-19 or AZD-1222, from the University of Oxford and AstraZeneca, which uses an incomplete chimpanzee adenovirus (ChAd) that incorporates the *S* gene with optimized codons [168]. The Sputnik-V vaccine, developed by Gamaleya Research Institute of Russia, combines the adenovirus vectors Ad26 and Ad5 that encode the S protein [169].

Among the vaccines based on recombinant proteins is the NVX-CoV2373 vaccine from the company Novavax, which uses the complete recombinant S protein in trimeric form expressed and purified in insect cells [170].

As inactivated vaccines for SARS-CoV-2, there are those developed by the Wuhan Institute of Biological Products [171] and by the Beijing Institute of Biological Products [172]. The PiCoVacc inactivated virus vaccine is a multivalent vaccine developed by Sinovac, which has been shown to generate antibodies with neutralizing activity for nine different SARS-CoV-2 isolates representative of the variants circulating worldwide [173].

Live attenuated vaccines include strains with lower virulence and that maintain a normal replicative capacity; like those developed by Chungnam National University [174]. All vaccines developed against COVID-19 had high efficacy against the original strain and the Alpha, Beta, Gamma, or Delta variants of concern and were well tolerated. The efficacy of the COVID-19 vaccine or its effectiveness against serious diseases is high, although it decreases six months after complete vaccination [162,175]. Vaccines developed to control COVID-19 have provided sufficient information that could be used to control other infections caused by coronaviruses.

Regarding PEDV vaccines, inactivated virus vaccines were developed in the 90’s in Asia [176]. However, these vaccines use strains that currently are classified as classic or ancestral strains, and they are not effective against the currently circulating strains [177]. Since the 2010 outbreak in China, attenuated virus vaccines based on the Zj08 strains [176], 83P-5 [177], and DR-13 [178] have been created. Since the 2013 outbreak, the development of PEDV vaccines started in the United States. Harrisvaccines was one of the pioneering companies that produced the iPED vaccine, based on the truncated version of the *S* gene using the SirraVax^TM^ RNA particle technology platform [179].

Subsequently, the same company created the second generation of the vaccine called iPED plus, which generates a longer fragment of the S protein with optimized codons. Other vaccines were those created by Zoetis, using inactivated virus [180], and the Vaccine and Infectious Disease Organization-InterVac in Canada, based on the recombinant production of the S1 domain of the S protein in human cells [181].

An alternative vaccine was generated in Lactobacillus expressing the N and S proteins of PEDV to stimulate the production of immunoglobulin A (IgA) and IgG specifically in the porcine intestinal mucosa [182,183].

DNA-based vaccines have also been developed, and such vaccines induce high levels of anti-PEDV antibodies in vaccinated animals [184]. Another DNA vaccine is called SL7207, expressing the S protein of PEDV and TGEV, and its inoculation in pigs demonstrated a high cellular and humoral immune response and the generation of neutralizing antibodies [185]. Another vaccine uses the recombinant adenovirus rAD-vDEP-S to induce a high humoral immune response in inoculated pigs and partial protection in pigs naturally infected with PEDV [186].

The emergence of SARS-CoV-2 has positioned mRNA technology as a new platform for vaccine development, however, its application in the control of PEDV is distant. On the other hand, the appearance of virulent PEDV strains with genetic material from attenuated strains that served as vaccines suggests the need to reevaluate the viability of the attenuated vaccines that are being used for SARS-CoV-2. In addition, based on the information from the PEDV vaccines, it is possible to suggest that SARS-CoV-2 vaccines produced with different technologies and platforms could have compromised effectiveness due to possible recombination or mutation events. Therefore, the production of vaccines on an annual or biannual basis is a possibility that should not be ruled out.

Finally, continuous monitoring of PEDV and SARS-CoV-2 variants is necessary, as well as evaluation of the effectiveness of existing vaccines against newly re-emerging strains.

## 8. Conclusions

PEDV significantly affects the pork industry and therefore the supply of pork in affected populations while SARS-CoV-2 affects human health and economic activities. The viruses share many of their characteristics in viral infection as well as in the induced cellular response. Some discoveries about SARS-CoV-2 had already been previously de-scribed in other coronaviruses. However, the scale of the COVID-19 pandemic has attracted diverse scientific disciplines, opening new possibilities for analysis and understanding, including for other coronaviruses such as PEDV. In countries affected by PEDV, establishing a “national coronavirus research network” is essential to promote scientific and economic resilience. Finally, staying abreast of current information on infections caused by different coronaviruses equips us to confront new challenges, whether they arise from emerging or re-emerging viruses.

## Figures and Tables

**Figure 1 viruses-16-00238-f001:**
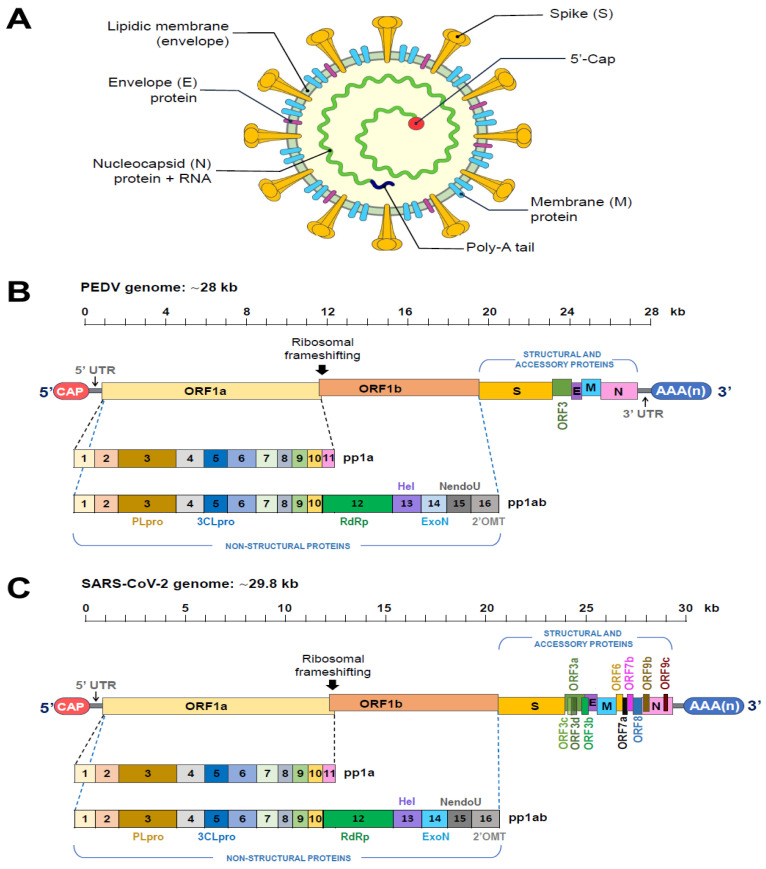
Schematic diagram of the virion structure of coronaviruses and the PEDV and SARS-CoV-2 genomes structure. PEDV and SARS-CoV-2 genome. (**A**). General structure of the virion. (**B**). PEDV genome organization. The viral genome is composed of seven open reading frames (ORFs) and the 5′ and 3′ untranslated regions (UTRs). The ORFs encode four structure proteins, spike (S), envelope (E), membrane (M), and nucleocapsid (N), an accessory protein (ORF3), and 16 non-structure proteins or NSPs (NSP1–16). NSP3 contains two papain-like protease domains (PLP1 and PLP2). Sites cleaved by proteases 3CLpro, 3C-like protease (NSP5). RdRp, RNA-dependent RNA polymerase (NSP12); HEL, 5′-to-3′ helicase (NSP13); ExoN, 3′-to-5′ exoribonuclease (nsp14); N7-MTase, N7-methyl transferase (NSP14); EndoU, endoribonuclease (NSP15); 2′-O-MTase, 2′-O-methyl transferase (NSP16). (**C**). SARS-CoV-2 genome encodes a total of 29 CoV-2 proteins including 16 non-structural proteins (NSP1–NSP16), 4 structural proteins (S, E, M, and N), and 9 accessory ORFs (3a, 3b, 6, 7a, 7b, 8, 9b, 9c, and 10). The two main transcriptional units, ORF1a and ORF1ab, encode replicase polyprotein 1a (PP1a) and polyprotein 1ab (PP1ab), respectively. The largest polyprotein PP1ab embeds non-structural proteins, which form the complex replicase machinery. The 5′ cap structure and 3′ poly (A)tail are for translation of ORF1a and ORF1ab that are further processed to generate sixtheen NSPs that include all major replicase genes and enzymes that are needed for subsequent viral transcription and replication. The cleavage sites of the proteases PLpro and 3CLpro are indicated.

**Table 1 viruses-16-00238-t001:** Comparison topics between PEDV and SARS-CoV-2.

Comparison Topics	PEDV	SARS-CoV-2
Genus	Alphacoronavirus	Betacoronavirus
Host	Pigs	Humans
Impact area	Pig industry	Economy and health
First description	1977 in England 1981 in Asia	2019 in Asia
Related viruses	TGEV	SARS, MERS
Disease produced	Porcine epidemic diarrhea	COVID-19
Symptoms	Vomiting, dehydration, watery diarrhea, weight loss	Fever, dry cough, respiratory distress, multiple organ failure in severe cases
Mortality	Up to 100% in piglets	High in people with underlying diseases (overweight, diabetes, cancer)
Transmission route	Oral–fecal route Distribution of particles by air Genetic material in the nasal cavity (without direct contact with the virus) Use of subepithelial CD3^+^ T cells to reach the intestine Detection in lung tissue (epithelial cells and alveolar macrophages) Infects dendritic cells (crosses the epithelial barrier to reach the intestinal mucosa; “Trojan horse”)	Person–person contact (airborne particles) Presence of particles in anal samples Possible transmission by fecal route Virus persistence in anal samples Symptoms of gastrointestinal diseases RNA samples in esophagus, stomach, and rectum CD3^+^
Viral receptor Cellular tropism	Protein aminopeptidase N (peptidases) Highly expressed in gastrointestinal tissues PEDV can infect APN-knockout cells and pigs APN promotes the entry of PEDV PEDV can infect human, monkey, bat cells Protein S requires proteolytic processing	ACE2 protein (peptidases) Highly expressed in intestine, testes, kidneys, heart, thyroid, eye tissue Average expression in colon, liver, bladder, and lungs High expression in bronchial transient secretory cells, nasal and bronchial epithelium, and in type II alveolar epithelial cells Uses the TMPRSS2 protease (highly expressed in intestines) ACE2 is highly expressed in alveolar type II cells of the lung and also in absorptive enterocytes of the ileum, colon, and stomach Protein S requires proteolytic processing
Classification	Genogroups G1 and G2 (G1a and b and G2a and b)	GISAID
Variants	PEDV S-INDEL and non-S-INDEL Hypervariable protein S	High frequency of mutations in protein S Nextstrain classifies by clades
Recombination	Recombination events between PEDV strains G2c classification Recombination between PEDV and TGEV Recombination between virulent and attenuated strains	Recombination events Recombinant viruses XA, XB, XC, XAA, XAB

**Table 2 viruses-16-00238-t002:** PEDV and SARS-CoV-2 vaccines.

	PEDV		SARS-CoV-2	
Technology Type	Company and/or Institution	Antigen	Company/Institution	Antigen
mRNA	Harrisvaccines	Truncated version of protein S	Pfizer and BioNTech Modern	RBD of S protein Full-length, prefusion stabilized S protein
DNA			Inovio Pharmaceuticals	S Protein
Recombinant protein	Vaccine and Infectious Disease	Organization-InterVac Domain S1	Novavax	S protein in trimeric form
Virus as a vector			CanSino Biological Inc. Janssen Vaccines & Prevention Oxford University and AstraZeneca Gamaleya Research Institute of Russia	S protein S protein S protein S protein
Inactivated virus	Zoetis	1 variant	Wuhan Institute of Biological Products Beijing Institute of Biological Products Sinovac	1 variant 1 variant Multivariate
Attenuated virus			Chungnam National University Duke-NUS Medical School of Singapore	

## Data Availability

Data are contained within the article.

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
