# Peer review of "Comparative Review of the State of the Art in Research on the Porcine Epidemic Diarrhea Virus and SARS-CoV-2, Scope of Knowledge between Coronaviruses"

_viruses, 2024, doi:10.3390/v16020238_

Round 1

Reviewer 1 Report

Comments and Suggestions for Authors

In this review, the authors aimed to consolidate the knowledge on PEDV and SARS-CoV-2, covering aspects such as viral genome, transmission, tropism, virus variants, vaccination, and immune response. The comparative analysis furnishes valuable insights into the distinctions and shared characteristics of these two coronaviruses. However, despite the comprehensive compilation of published information related to PEDV and SARS-CoV-2 research, the article lacks sufficient guidance for future research based on this comparative knowledge. Additionally, the readability of the article is compromised by lengthy paragraphs that need to be broken down. Each section follows a simple layout, allocating half for PEDV and half for SARS-CoV-2, but the author should engage in more proofreading and editing to enhance overall readability. The overall quality of this review needs to be improved before it can be published.

Comments on the Quality of English Language

The article requires meticulous proofreading to rectify typos, and further editorial efforts are necessary to enhance the overall flow of writing and organization.

Author Response

In this review, the authors aimed to consolidate the knowledge on PEDV and SARS-CoV-2, covering aspects such as viral genome, transmission, tropism, virus variants, vaccination, and immune response. The comparative analysis furnishes valuable insights into the distinctions and shared characteristics of these two coronaviruses. However, despite the comprehensive compilation of published information related to PEDV and SARS-CoV-2 research, the article lacks sufficient guidance for future research based on this comparative knowledge.

Work was done on the readability of the entire manuscript. Ideas have been better presented in relation to the orientation for future research based on comparative knowledge. Review considers the possibility of using the information obtained in the study of SARS-CoV-2 in the study of PEDV or other coronaviruses. Figure 1 was modified to improve the presentation of the information; a diagram of the general structure of the virion was included. At the end of each comparative topic addressed in the review, a statement is made regarding the scope of information, especially from SARS-CoV-2 to PEDV. For example,

Page 6. Lines 169-172. more research is required to understand in detail whether SARS-CoV-2 uses mechanisms like PEDV to reach intestinal tissue. This knowledge would be useful to propose new strategies to control infection by both PEDV and SARS-CoV-2.

Page 9. Lines 327-329. The classification systems of SARS-CoV-2 could be used for the phylogenetic classification of PEDV and other coronaviruses, which would facilitate almost real-time monitoring of the genetic variants.

Additionally, the readability of the article is compromised by lengthy paragraphs that need to be broken down.

Work was done on the readability of the article, the paragraphs have been broken down in order to improve the presentation of the information. The suggestion significantly improved the presentation of information.

Each section follows a simple layout, allocating half for PEDV and half for SARS-CoV-2, but the author should engage in more proofreading and editing to enhance overall readability.

The authors improved the readability of the manuscript so that the information presented is clear. Regarding the design of the structure of the manuscript, the authors considered that it was necessary to maintain the segmentation of the information, assigning half to PEDV and the other half to SARS-CoV-2, as in accordance with the original idea of the manuscript.

The overall quality of this review needs to be improved before it can be published.

The overall quality of the review was substantially improved, with significant work being done on the readability of the information and the review of the English language.

The article requires meticulous proofreading to rectify typos, and further editorial efforts are necessary to enhance the overall flow of writing and organization.

The observation was addressed, work was done to rectify typographical errors, the readability of the information was improved, and although the organization of the review was maintained, the general flow of the document was improved.

Reviewer 2 Report

Comments and Suggestions for Authors

This study is very interesting, to reveal the relation between PED virus infection and SARS-CoV-2 virus infection.

But I think it might be better to be revised bellow.

Introduction

I am not sure, you described that the first article published on PEDV appeared in 1981. But the disease of PED has been reported in the European and Asian pig industries over the last 30 years, with the virus first appearing in England (Wood, 1977) and Belgium (Pensaert and de Bouck, 1978) in the late 1970s. Please confirm this difference.

In the line 136 to 140, However, it does not replicate in this cell type, apparently PEDV uses dendritic cells to cross the epithelial barrier of the nasal cavity, phenomenon that has been identified in several viruses that take advantage of them as "Trojan horse" to overcome the epithelial barrier, evade antiviral immune responses and spread to the submucosal layer.

The word, Trojan horse is very unique but the meaning of this sentence is obscure. Please explain clearly this word “Trojan horse”.

If it would be possible, please add the schema of the same points of PED infection and SARS-CoV-2 virus infection. 

Author Response

The first comment for the reviewer is that the authors work on the readability of the entire manuscript.

I am not sure, you described that the first article published on PEDV appeared in 1981. But the disease of PED has been reported in the European and Asian pig industries over the last 30 years, with the virus first appearing in England (Wood, 1977) and Belgium (Pensaert and de Bouck, 1978) in the late 1970s. Please confirm this difference.

Yes, the comment is correct. The first description was in 1971, the first PEDV report took place in 1977 and the first report in Asia was in 1981 which was not clarified in table 1. It has been corrected.

In the line 136 to 140, However, it does not replicate in this cell type, apparently PEDV uses dendritic cells to cross the epithelial barrier of the nasal cavity, phenomenon that has been identified in several viruses that take advantage of them as "Trojan horse" to overcome the epithelial barrier, evade antiviral immune responses and spread to the submucosal layer. The word, Trojan horse is very unique but the meaning of this sentence is obscure. Please explain clearly this word “Trojan horse”.

Yes, the reviewer is right, for our working group the statement seemed clear but we did not state it correctly. Since we worked on the readability of the entire document, this paragraph was modified by eliminating the word Trojan horse.  The paragraph is organized as follows: Overcoming the epithelial/endothelial barrier could be a shared characteristic between coronaviruses. Apparently PEDV uses dendritic cells to cross the epithelial barrier, evade the antiviral immune response and reach the submucosal layer [33-34].

If it would be possible, please add the schema of the same points of PED infection and SARS-CoV-2 virus infection

We addressed this suggestion a little differently than what you kindly requested. We considered that it was necessary to improve figure 1 and not include another scheme. The first figure was intended to show the similarities in the structure of the virion and the organization of the genome between both viruses, we were not clear in the first shipment. The modification to the figure was minor but we believe it supports the presentation of information throughout the manuscript.